# Targeting the Warburg Effect in Cancer: Where Do We Stand?

**DOI:** 10.3390/ijms25063142

**Published:** 2024-03-08

**Authors:** Ignasi Barba, Laura Carrillo-Bosch, Joan Seoane

**Affiliations:** 1Faculty of Medicine, University of Vic-Central University of Catalonia, 08500 Vic, Catalonia, Spain; 2Vall d’Hebron Institute of Oncology (VHIO), CIBERONC, Vall d’Hebron University Hospital, Universitat Autònoma de Barcelona, 08035 Barcelona, Spain; 3Institució Catalana de Recerca i Estudis Avançats (ICREA), 08010 Barcelona, Spain

**Keywords:** tumor metabolism, tumor microenvironment, Warburg effect, aerobic glycolysis, immunomodulation

## Abstract

The Warburg effect, characterized by the preferential conversion of glucose to lactate even in the presence of oxygen and functional mitochondria, is a prominent metabolic hallmark of cancer cells and has emerged as a promising therapeutic target for cancer therapy. Elevated lactate levels and acidic pH within the tumor microenvironment (TME) resulting from glycolytic profoundly impact various cellular populations, including macrophage reprogramming and impairment of T-cell functionality. Altogether, the Warburg effect has been shown to promote tumor progression and immunosuppression through multiple mechanisms. This review provides an overview of the current understanding of the Warburg effect in cancer and its implications. We summarize recent pharmacological strategies aimed at targeting glycolytic enzymes, highlighting the challenges encountered in achieving therapeutic efficacy. Additionally, we examine the utility of the Warburg effect as an early diagnostic tool. Finally, we discuss the multifaceted roles of lactate within the TME, emphasizing its potential as a therapeutic target to disrupt metabolic interactions between tumor and immune cells, thereby enhancing anti-tumor immunity.

## 1. Introduction

Almost a century ago, Otto Warburg described that tumors undergo the oxidation of glucose into lactic acid, even in the presence of oxygen [1]. This phenomenon, now known as the Warburg effect or aerobic glycolysis, involves the conversion of glucose into pyruvate and subsequently into lactate, despite cancer cells possessing a fully functional mitochondrial respiratory chain and complete biochemical competence to utilize this pathway for ATP production.

The metabolic shift towards the oxidation of glucose into lactate stands out as a distinctive feature across various tumor types. What is more, altered cellular metabolism has been identified as one of the characteristic hallmarks of cancer [2]. Thus, the prevalence of a high rate of glucose catabolism into lactate is the most widespread metabolic phenotype observed in cancer cells [3]. The presence of the Warburg effect has been described in most tumor types, including, but not limited to, glioblastoma [4], pancreas [5], breast [6,7], and cervix [8].

## 2. Glycolysis

Glucose enters the cell via glucose transporters (GLUT), a group of transmembrane proteins that allow glucose entrance by facilitated diffusion (Figure 1). Once within the cell, glucose undergoes phosphorylation to form glucose-6-phosphate, catalyzed by hexokinase (HK). This reaction consumes one ATP molecule, rendering the glucose molecule charged and effectively trapped inside the cell. Glucose-6-phosphate can go into the pentose phosphate pathway (PPP) or enter glycolysis, where it undergoes a series of conversions through hexose phosphates, ultimately producing fructose 1,6-bisphosphate. This transformation requires an additional ATP molecule and is catalyzed by phosphofructokinase (PFK). HK and PFK serve as key regulatory enzymes in glycolysis.

Fructose 1,6-bisphosphate is then cleaved into two 3-carbon sugars, dihydroxyacetone phosphate and glyceraldehyde 3-phosphate, which are biochemically interconvertible. Glyceraldehyde 3-phosphate undergoes phosphorylation with inorganic phosphate to yield 1,3-bisphosphoglycerate. The subsequent oxidation of this molecule to pyruvate results in the generation of two ATP and one NADH molecule. However, since two glyceraldehyde 3-phosphate molecules are produced for each glucose molecule, the net yield of the glucose-to-pyruvate oxidation is two ATP and two NADH.

In cells with fully functional mitochondria, NADH fuels the electron transport chain. However, cancer cells must decrease their NADH pool through alternative means, as it acts as an allosteric inhibitor of, among others, PFK, potentially limiting the glycolytic rate at high levels. In this context, NADH is utilized to reduce pyruvate to lactate, a reaction catalyzed by lactate dehydrogenase (LDH). Excess lactate is then exported to the extracellular tumor microenvironment (TME) via monocarboxylate transporters (MCT), where it can accumulate up to a concentration of 40 mMol/L and reduce the pH to between 6 and 7 [9].

## 3. Why Warburg Effect?—Mechanism and Advantages for Cancer Cells

At the time of its discovery, tumoral metabolic reprogramming was thought to be the driving force behind tumor transformation [1]. Later, it was assumed that metabolic alterations are not the cause but rather a consequence of the tumorigenic process, while contemporary understanding asserts that there is an interconnection between genes and metabolism. Several oncogenes, including *MYC*, *KRAS*, *Wnt*, and others, are implicated in inducing metabolic changes [10,11,12,13]. This view is supported by the fact that therapeutic approaches targeting oncogenic cellular signaling may impact tumor metabolism, as exemplified by c-*Myc* inhibition leading to the concurrent activation of Peroxisome Proliferator-Activated Receptor Alpha signaling [14]. However, it has also been suggested that mitochondria can sense metabolic changes and send information to the nucleus, a process known as mitochondrial reverse signaling or mitochondrial retrograde signaling [15]. Mitochondrial genetic stress induced by depletion of mtDNA is associated with increased markers of tumor invasion, activation of ERK kinases, and increased levels of Bcl2 in a lung carcinoma cell line [16]. Cancer cells can activate the mitochondrial reverse signaling in response to mitochondrial dysfunction, such as diminishing ATP levels or reduced membrane potential; these signal pathways subsequently promote tumor growth and progression [17]. 

While the oxidation of glucose to pyruvate yields a modest amount of two ATP molecules, the complete oxidation of glucose to CO_2_ and the generation of energy through oxidative phosphorylation (OXPHOS) can yield between 31 and 38 ATP molecules, depending on the NADH shuttle employed [18] and whether the yield of NADH/FADH_2_ is considered as an integral or non-integral number [19]. Despite the inefficiency of aerobic glycolysis compared to oxidative phosphorylation, tumors and other fast-dividing cells that use the Warburg effect for ATP production maintain intracellular ATP levels comparable to non-tumoral cells [1].

To elucidate why cancer cells adopt the Warburg effect despite its apparent energetic disadvantage, it has been hypothesized that “the metabolism of cancer cells, and indeed all proliferating cells, is adapted to facilitate the uptake and incorporation of nutrients into the biomass needed to produce a new cell” [20]. In this context, aerobic glycolysis serves to furnish cells with essential biosynthetic intermediates, such as ribose for nucleotide synthesis and glycerol and citrate for lipid synthesis, as well as nonessential amino acids. Notably, the PPP assumes significance, providing precursors for nucleotide, amino acids, and vitamin B6 biosynthesis while also playing a key role in the regulation of oxidative stress through NADPH-mediated reduction of glutathione [21,22]. Recent in vivo evidence supports the activation of the PPP in highly glycolytic tumors where the Warburg effect is prominent [23].

Cachexia manifests as a multiorgan disorder characterized by energy imbalance and involuntary loss of lean body mass with or without reduction of adipose tissue. It affects nearly half of cancer patients and is strongly correlated with poor prognosis. One important mechanism underlying cachexia involves the heightened resting energy expenditure driven by the increased activation of futile metabolic cycles, like the Cori cycle. This process facilitates the conversion of lactate to pyruvate and subsequently to glucose in the liver, resulting in elevated energy expenditure. The glucose is then transported into circulation and back to tumor tissues, where it sustains high glucose consumption and is converted back into lactate. The Cori cycle has been described to account for 50% of glucose turnover in cachectic cancer patients, compared to 20% in cancer patients without this condition [24]. In a study involving pancreaticobiliary adenocarcinoma patients, researchers demonstrated a correlation between cachexia and a high glycolytic index as measured by the expression levels of various glycolytic enzymes [5]. Furthermore, the same authors also show in a nude mice model that inhibiting the Warburg effect does attenuate cachexia [25], suggesting a potential causal link between the Warburg effect and cancer-induced cachexia. However, other factors involved in cancer-associated cachexia, such as systemic inflammation or anorexia, have also been reported to play a significant role in this complex pathology and are extensively reviewed elsewhere [26,27].

### Warburg Effect Impact on the Tumor Microenvironment

The TME constitutes a complex and dynamic system that comprises cancer cells, stromal tissue (immune cells, fibroblasts, and vascular tissue), signaling molecules (cytokines), and the extracellular matrix [28]. TME can actively promote cancer progression and weak tumor immunosurveillance [29]. Importantly, TME has a significant impact on immunotherapy response [30] and modulates the response to chemo and radiotherapy [31].

One of the main characteristics of the TME is an increase of lactate and acidification of the TME to pH between 6.0 and 6.5 as a result of the Warburg effect [32]. Notably, MCT4, a lactate exporter, remains functional even in highly acidic, high-lactate environments [33], ensuring uninterrupted glycolysis even in such challenging conditions. Metastatic tumors exhibit higher lactate levels in the TME compared to non-metastatic ones, with lactate levels correlating with decreased survival rates in cervix and head and neck tumors [34]. 

Both lactate and acidification exert profound effects on the phenotype and functionality of the diverse populations within the TME and on the tumoral cell capacities (Figure 2). Lactate, apart from being one of the most abundant metabolites in the TME, can act as a signaling molecule through multiple receptors [35]. It has been reported to promote tumor metastasis through the activation of the NF-κβ pathway in tumor cells [36] or the PI3K-AKT-CREB pathway [37]. Acidosis can also induce a cancer-specific signaling cascade, including NF-κβ, facilitating cell invasion [38]. Furthermore, the elevated lactate levels and acidic pH play a crucial role in the interaction between tumor and non-tumor cells within TME [39], promoting immunosuppressive phenotypes in both innate immune cells [40] and adaptive immune cells [41]. 

Metabolic characteristics in the microenvironment control macrophage phenotypes and functions [42]. Tumor-derived lactic acid is able to polarize tumor-associated macrophages (TAMs) towards an anti-inflammatory/immune-suppressive phenotype, thereby promoting tumor maintenance and growth [43]. Conversely, macrophage polarization can trigger metabolic shifts in the macrophages [42]. In oral squamous cell carcinoma, a positive feedback loop between tumoral cells and macrophages has been observed in which tumor-derived lactic acid induces the synthesis of glycoprotein non-metastatic protein B in macrophages and that, in turn, acts as a paracrine molecule facilitating tumor migration and invasion [44]. Macrophages also possess the ability to suppress T-cell recruitment and regulate other aspects of tumor immunity [45]. High lactate levels also prevent monocytes from differentiating into mature dendritic cells and promote a tolerogenic phenotype, leading to the secretion of immunosuppressive cytokines [46]. Additionally, T-cells and natural killer (NK) cell functionality is impaired by lactic acidosis. Activated T-cells, which heavily rely on glycolytic metabolism, are adversely affected in the TME due to low glucose and high lactate concentrations, compromising their anti-tumor activity. Furthermore, extracellular acidification suppresses CD8+ T-cell functionality through p38/JNK pathway inhibition and reduces INFy production and secretion [47]. Lactic acidosis also suppresses the anti-tumoral activity of NK cells through the mTOR pathway inhibition [48]. However, regulatory T-cells have been shown to be highly resistant to increased extracellular lactate levels and low pH, maintaining their survival and immunosuppressive role by upregulating the expression of FOXP3 [49]. Lactic acidification of the TME diminishes immunosurveillance and promotes a favorable environment for tumor progression. 

Cancer-associated fibroblasts (CAFs) exhibit increased glycolysis and enhanced export of lactate to the microenvironment, a process known as the reverse Warburg effect [50]. Evidence of a lactate shuttle between CAFs and cancer cells has been demonstrated in prostate cancer [51], allowing the use of CAFs to produce lactate to fuel the tumor through the tricarboxylic acid cycle in order to maintain the high energetic demand. Moreover, CAFs secreted lactate exerts immunosuppressive activity in the TME of pancreatic cancer [52]. This dual role of lactate as an energy source and signaling molecule highlights the complexity of cellular and metabolic interactions within the TME. 

## 4. Therapeutic Strategies Targeting the Warburg Effect

Therapeutic strategies targeting the Warburg effect have gained attention in the scientific community in recent times [53,54,55] Table 1. Significantly, fasting blood glucose levels have been correlated with prognosis, suggesting that glucose availability to the tumor may influence survival [56]. Furthermore, a ketogenic diet, coupled with calorie restriction, has demonstrated efficacy in impairing tumor growth in animal models of brain tumors [57]. Phase I clinical trials have been conducted with a ketogenic diet in cancer patients, but definitive evidence of its effectiveness is still lacking. Nevertheless, these trials have indicated that a ketogenic diet can be implemented in clinical settings with minimal side effects [58], paving the way for larger trials aimed at establishing its efficacy.

### 4.1. GLUT Transporters

The GLUT transporters, a family of transmembrane proteins facilitating glucose transportation across the cellular membrane, mediate the initial step in cellular glucose utilization. High expression of glucose transporters GLUT1 and/or GLUT3 is commonly associated with poor prognosis in several cancer types, including papillary thyroid carcinoma and colorectal cancer [78,79]. Notably, GLUT2 is overexpressed in hepatocellular carcinoma [78]. Regardless of the specific transporter involved, it is crucial to recognize that glucose uptake serves as a rate-limiting step for hyperproliferation, as highlighted already by Warburg in 1956 [1]. 

Considering these observations, GLUT transporter inhibition has been proposed as a crucial therapeutic target of aerobic glycolysis [80,81]. Blocking glucose uptake also impacts the PPP and hinders NADPH production, thereby limiting the tumor’s antioxidant defense [61].

Recent studies have reported new small molecules targeting GLUTs, such as KL-11743, which has demonstrated the ability to inhibit glucose uptake in vivo [61]. Other GLUT inhibitors, such as BAY-876, exhibit the capacity to impede the growth of triple-negative breast cancer in patient-derived xenograft models [82]. Additionally, Glutor, a pan-GLUT inhibitor, has also been reported to exert antineoplastic effects [59]. Notably, the inhibition of GLUT1 induces cell death, overcoming resistance to chemotherapeutic agents in cultured gastrointestinal cell lines [83] and sensitizing radio-resistant breast cancer cells [84].

### 4.2. Hexokinase

Upon entry into the cell, glucose undergoes phosphorylation catalyzed by HK, a key enzyme that irreversibly transfers a phosphate group from ATP to the glucose molecule (except in some specialized cells of the liver and kidneys that have glucose-6-phosphatase activity). This phosphorylation renders glucose charged and confines it within the cell, committing it to either glycolysis or the PPP. Thus, HK plays a pivotal role as a regulator in the fate of glucose metabolism. The expression of HKII isoform is linked to the proliferation of hepatoma cells [85], and its overexpression correlates with poor prognosis in various tumor types, including tumors of the digestive system [82], brain metastases of breast cancer [86], and is indicative of a poor therapeutic response in circulating lung tumor cells expressing high levels of HKII [87].

A classic inhibitor of HK is 3-bromopyruvate (3BP), which has shown promising results in pre-clinical studies either as a single therapeutic agent or in combination with other anti-tumoral drugs such as tamoxifen [88]. Different encapsulation strategies for drug delivery aimed at reducing side effects have also demonstrated success in a pancreatic cancer model [89]. However, the use of 3BP in humans is limited, with only two case reports published to date [62,63]. A comprehensive review of 3BP as an anticancer agent is available in the work by Fan and colleagues [90].

2-Deoxyglucose (DOG), a glucose analog, enters the cell through GLUT transporters and is also phosphorylated by HK. However, the resultant DOG-phosphate accumulates in the cell since it cannot be used for further steps of the glycolytic pathway and cannot be excreted due to its electrically charged nature. Thus, it effectively inhibits the phosphorylation of new glucose molecules by HK. Despite early anti-tumoral promising effects in human studies, clinical trials were discontinued due to side effects compatible with hypoglycemia symptoms among participants, even though they had normal blood glucose levels [64]. This observation can be explained by the inability of circulating glucose to enter the cells due to the accumulation of DOG-phosphate. Despite these early setbacks, the use of DOG in combination with chemotherapy [91] or radiotherapy [92] has continued into the 21st century, proving feasible with doses of 68 and 200 mg/kg, respectively, without serious side effects.

Moreover, a novel orally available HK inhibitor, Benitrobenrazide, has demonstrated the ability to block glycolysis and inhibit cancer cell growth in a mouse xenograft model without apparent side effects [93].

### 4.3. Glyceraldehyde 3-Phosphate Dehydrogenase (GAPDH)

GAPDH catalyzes the simultaneous phosphorylation and oxidation of glycerol-3-phosphate to 1,3-biphosphoglycerate, utilizing NAD^+^ as the electron acceptor, in a reaction that is reversible under physiological conditions. However, emerging evidence suggests that GAPDH is a multifunctional protein with a significant role in regulating cell death [94]. Recent research has implicated GAPDH in the regulation of glycolysis within the Warburg effect [95].

The selective inhibition of GAPDH with koningic acid, a fungal metabolite also known as heptelidic acid, has demonstrated the ability to reduce glycolysis and inhibit tumor growth in sensitive animals [65]. Orally administered koningic acid has been shown to effectively inhibit GAPDH in transplanted skin cancer cells in mice [96]. However, despite these promising findings, the use of GAPDH inhibitors has not yet reached clinical trials.

### 4.4. Triose Phosphate Isomerase

The primary role of triose phosphate isomerase (TPI) is to catalyze the reversible interconversion of dihydroxyacetone phosphate (DHAP) and glyceraldehyde-3-phosphate (G3P). This reaction is close to the equilibrium and has not traditionally been considered a major therapeutic target. However, TPI has been found to accumulate in various cancer types [97] and has the capability to activate signaling pathways such as PI3K/AKT/mTOR [98] and MAPK/ERK [99]. Of particular interest is the observation that deamidated TPI accumulates in breast cancer cells but not in healthy cells [66]. Deamidation is a spontaneous post-transcriptional spontaneous modification of proteins consisting of asparagine deamidation into aspartic acid and isoaspartate. The pharmacological inhibition of deamidated TPI, while sparing unmodified TPI, is an effective way of selectively targeting tumoral cells, and this approach has been shown to result in tumor size reduction in an in vivo mice model [66].

### 4.5. Phosphofructokinase

The allosteric regulation of glycolysis by PFK enables cancer cells to modulate their glycolytic flux precisely, addressing both bioenergetic and biosynthetic demands [100]. Often referred to as the gatekeeper of glycolysis [101], PFK is highly expressed in many tumor types, exerting regulatory control over cancer cell growth and metabolism [102,103,104]. In recent years, the development of new PFK inhibitors, including 3PO [69], PFK15 [70], and PFK158 [71], has shown promising results in pre-clinical models

Despite the potential, a phase I clinical trial using PFK158 was discontinued due to limited success [72]. The challenges may arise from the multifunctional role of PFK in various cellular processes [100], possibly inducing side effects upon targeting. Also, PFK has been found to be a driver of bevacizumab resistance through non-metabolic processes [105]. Nevertheless, new structural insights provide hope for the rational development of novel PFK glycolytic inhibitors [101].

### 4.6. Phosphoglycerate Mutase

Phosphoglycerate mutase (PGAM1) catalyzes the internal transfer of a phosphate group from C-3 to C-2, which results in the conversion of 3-phosphoglycerate to 2-phosphoglycerate through a 2,3-bisphosphoglycerate intermediate. PGAM1 was found to promote tumor growth through the coordination of glycolysis and biosynthesis [67]. Additionally, PGAM1 exhibits non-metabolic functions; for instance, it has been observed to facilitate actin filament assembly, cell motility, and cancer cell migration [106]. The small molecule HKB99 allosterically blocks PGAM1 and has demonstrated the ability to reduce tumor growth and metastatic potential in a mice model of non-small cell lung cancer by ROS-dependent activation of JNK/c-Jun signaling and abrogation of PGAM1 and ACTA2 interaction [68].

### 4.7. Lactate Dehydrogenase

Pyruvate serves as the end product of aerobic glycolysis. However, for the Warburg effect to persist, the two NADH molecules produced during glucose oxidation to pyruvate must be oxidized, as they act as allosteric inhibitors of PFK and obstruct glycolysis. LDH plays a crucial role by reducing pyruvate to lactate, coupled with the oxidation of NADH to NAD. The excess lactate is then transported out of the cell via MCTs.

LDH, like other glycolytic enzymes, is overexpressed in various tumor types, where high levels of LDH expression are associated with a poorer prognosis [107]. Additionally, the downregulation of LDHA expression correlates with decreased tumorigenicity [108]. Pharmacological inhibition of LDH with the small molecule FX11 reduces ATP levels and increases oxidative stress, resulting in the inhibition of tumor xerograph progression [73]. Other inhibitors, such as NHI-Glc-2, exhibit anti-tumor activity alone and have synergetic effects with gemcitabine in mice bearing tumors [74]. NCI-006, another LDH inhibitor, is also able to block tumor growth in mice, and the inhibition increases when it is used in combination with metformin [75]. Consequently, LDH inhibition is considered a promising target against breast cancer [109].

### 4.8. Monocarboxylate Transporters

In the context of the Warburg effect, the continual removal of the end product lactate from the intracellular space is crucial for the progression of biochemical reactions. Its secretion is facilitated by the MCT family of proteins [110,111]. Among the most relevant members for lactate transport are MCT1 (*Slc16a1*) and MCT4 (*Slc16a3*), associated with lactate import and export from the cell, respectively [110]. MCT4, functioning as a symporter with H^+^, not only exports lactate but also lowers extracellular pH [112]. Importantly, MCT4 maintains its ability to export lactate/H^+^ even in the challenging conditions of high lactate and low pH found in the TME [33].

Similar to other enzymes and transporters involved in the Warburg effect, MCTs are overexpressed in tumors as compared to normal tissue [113]. Inhibition of MCTs has been shown to induce cancer cell death [114]. Notably, the inhibition of MCT1 with AZD3965 has progressed to phase I clinical trials, where it was shown to be well tolerated at doses that effectively inhibit MCTs [76]. Furthermore, recent research has revealed that VB124, a specific MCT4 Inhibitor, demonstrates effectiveness in animal models [77].

## 5. Warburg Effect Targeting Resistance Mechanisms

Unfortunately, even after many years of research, advanced drug development, and success at the pre-clinical stage, therapies targeting glycolysis enzymes responsible for the Warburg effect have not yet reached widespread clinical practice. Several reasons can explain this lack of translation to the clinic. First, the extensive use of glycolysis by multiple tissues in the human body, not just tumoral or fast-growing cells, introduces the challenge of potential side effects. For instance, the clinical trial involving DOG was abandoned due to hypoglycemia-like symptoms despite normal blood glucose levels, highlighting the intricate impact of disrupting glycolysis on cellular function [64]. However, recent findings show that post-transcriptionally modified TPI tends to accumulate in cancer cells and can be selectively targeted [66], which may open up the possibility of disrupting cancer cell metabolism without severe side effects.

Targeting LDH and MCTs, while potentially having less impact on non-tumoral cells, still presents challenges. In principle, non-tumoral cells should be able to channel excess NADH to the mitochondria, allowing for a functional PPP to balance their redox stress while ATP production would continue through their functional mitochondria. However, the full blockade of LDH activity in a genetic model resulted in anemia after a few weeks of LDH knock-out induction [115]. Erythrocytes rely solely on aerobic glycolysis as they lack mitochondria; even though their metabolic requirements are low, a full blockade of LDH activity in time is clearly deleterious. 

The main regulatory steps of glycolysis, including HK [116], PFK [100], and LDH [117], are also controlled by enzymes that present different isoforms that have non-metabolic functions on top of catalyzing reactions of the glycolytic pathway [118]. For example, HKII is being proposed as an enzyme linking metabolic and survival pathways [119]. The C-terminal domain of PFKFB3 variant 5 localizes the enzyme to the nucleus, where fructose-2,6-bisphosphate increases the expression and activity of cyclin-dependent kinase-1, promoting cell proliferation [120]. Consequently, the possibility of inducing side effects associated with the inhibition of glycolytic enzymes cannot be discarded.

Additionally, the existence of redundant enzymes and transporters, such as various members of the GLUT family and MCTs, makes the complete inhibition of the glycolytic pathway difficult. Blocking a single isoform or transporter, as demonstrated with GLUT1 inhibition using BAY-876, may reduce the rate of lactate production but does not entirely halt glycolysis due to compensatory mechanisms [60]. Moreover, it has been seen that intracellular lactate increases only if both MCT4 and MCT1 are blocked [77]. Similarly, genetic disruption of both LDH A and B is required to ablate aerobic glycolysis [121].

Oshima and colleagues showed that cancer cells had the ability to shift from aerobic glycolysis to oxidative phosphorylation (OXPHOS) upon LDH inhibition; consequently, the combination of LDH and respiratory transport chain complex I inhibition prolonged the therapeutic benefits [75]. The identification of alternative energy/building block sources, such as acetate metabolism in some tumors [122,123], emphasizes the complexity and heterogeneity of cancer metabolism. 

As with other aspects of tumor biology, the metabolic landscape is dynamic and context-specific, playing a crucial role in cancer progression. Metabolic heterogeneity adds an additional layer of complexity, influencing differences in metastatic potential [124]. This evolving understanding is essential for refining therapeutic approaches and overcoming the challenges associated with targeting glycolysis in cancer treatment.

## 6. Cancer Detection through the Warburg Effect

Although targeting the Warburg effect has not become standard clinical practice, the heightened glucose oxidation observed in many tumors has found utility in cancer detection and assessing therapeutic response through positron emission tomography (PET) scanning [125,126]. PET scanning, often coupled with computed tomography, is widely employed in oncology [127] and it is invaluable in cancer staging, therapeutic planning, and response assessment [128].

The primary PET imaging tracer, ^18^F-FDG, incorporates a positron emitter (^18^F) bound to a DOG molecule. Once inside cells, DOG is phosphorylated by HK, trapping the tracer. A strong PET signal, historically linked to the Warburg effect, reflects elevated glucose uptake but not necessarily its conversion to lactate [129]. However, highly metabolic tissues like the brain may appear hyperintense in PET scans, and not all tumors are detectable through this imaging technique. Also, pathologies other than cancer, such as inflammation, may also manifest in PET scans [127]. Despite these limitations, a positive correlation between GLUT1 expression and ^18^F-DOG signal has been described in various tumor types [80].

The Warburg effect can also be visualized in vivo using hyperpolarized ^13^C MRI [130]. Rodrigues and colleagues [23] followed the fate of hyperpolarized [U-^2^H, U-^13^C]glucose in tumor-bearing mice and observed labeled lactate only in tumors. Metabolic changes detected as early as 24 h after treatment initiation precede alterations in tumor volume assessed by standard imaging techniques [131]. Hyperpolarized ^13^C MRI presents some advantages in comparison to PET. It is a non-radioactive and highly sensitive technique. However, the short half-life of hyperpolarized compounds used in ^13^C MRI poses a challenge for widespread clinical adoption. Despite this limitation, successful applications in prostate cancer [132] and breast cancer patients [133] underscore its clinical feasibility. Metabolism holds significant potential for diagnosis and response to treatment evaluation [134]. The ability of hyperpolarized ^13^C MRI to capture metabolic changes upon treatment process positions it as a valuable tool for assessing therapeutic effectiveness early in time.

## 7. Future Perspectives

In recent years, various small molecule inhibitors targeting glycolysis enzymes have emerged, showing promising results in both pre-clinical [69,70,71,77] and clinical trials [76], highlighting the therapeutic interest of the Warburg effect. The acidification of the extracellular medium and relative basification of the cellular cytoplasm induces chemoresistance by neutralizing weakly basic drugs such as paclitaxel, making it difficult for them to cross the membrane [31]. Furthermore, multi-drug resistance mediated by the export of drugs through P-glycoprotein could also be influenced by pH alterations [135]. Therefore, targeting the Warburg effect could offer significant benefits by reducing chemo- and radio-resistance. In this regard, clinical trials involving the use of DOG in combination with radiotherapy [92] or chemotherapy [91] have shown good tolerance to the combined treatment in the relatively low number of patients included in both trials.

Numerous authors have highlighted the potential of inhibiting glycolytic enzymes involved in the Warburg effect in enhancing the effects of immunotherapy [136]. For instance, inhibition of HK with clotrimazole reduces extracellular lactate, thereby potentiating anti-tumor immunity [137]. Similarly, Wegiel’s group demonstrated that LDH deletion triggers anti-tumor immunity, proposing LDH-A inhibitors as a strategy to enhance checkpoint inhibitor efficacy [138]. Moreover, genetic blockade of LDH has shown an additive effect on anti-PD1 therapy, suggesting modulation of LDH/lactate as a means to improve anti-PD1 treatments [139]. Additionally, blocking lactate export through MCT1, using AZD3965 [140], or inhibiting both MCT1 and MCT4 with diclofenac [141] preserves T-cell function and enhances immunotherapy response. 

Lactate, the Warburg effect’s end-product, plays a role as an immunosuppressor in the TME [43], influencing immunotherapy resistance through metabolic crosstalk between tumor and immune cells [142,143]. In this regard, it has been described that rewiring glucose metabolism in macrophages allows anti-tumor activity, including engulfment of CD47+ cancer cells. Thus, carbon metabolism has been proposed as a potential therapeutic target for stimulating anti-tumor activity by macrophages [144]. Furthermore, tuning tumor-associated macrophages (TAMs) into immunostimulatory macrophages also promotes T-cell response and has been described as a promising therapeutic strategy [145]. Consequently, the efficacy of treatments targeting the Warburg effect may be attributed, at least in part, to their impact on the TME. However, activated macrophages rely on aerobic glycolysis and it has been shown that the inhibition of GAPDH by 4-octyl itaconate downregulates aerobic glycolysis in activated macrophages, shifting its phenotype towards anti-inflammatory [146]. Thus, while inhibiting the Warburg effect does facilitate immune activation against tumors by modulating the TME, it may also have a direct negative effect when acting on macrophages.

## 8. Conclusions

The Warburg effect, a distinctive feature of cancer metabolism, not only provides valuable insights into cancer diagnostics through advanced imaging techniques but also stands out as a promising target for therapeutic interventions. The development of small molecule inhibitors targeting cancer-specific mutations in key glycolytic enzymes demonstrated promising results in pre-clinical studies and early clinical trials, although they have not yet reached extensive clinical use. However, new insights into the understanding of the Warburg effect microenvironment open new avenues for combinatory approaches, including those aimed at reducing the amount of lactate in the TME in order to enhance the effectiveness of immunotherapy.

## Figures and Tables

**Figure 1 ijms-25-03142-f001:**
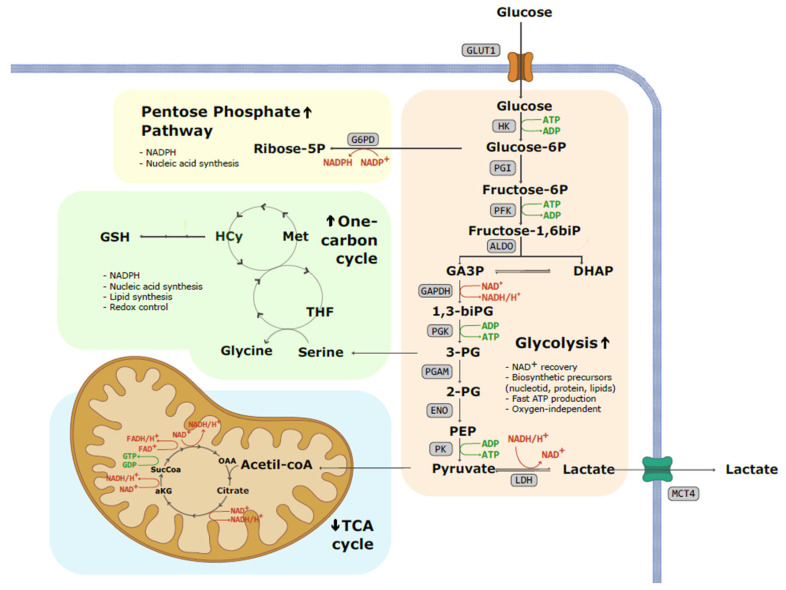
Schematic representation of the metabolic fate of glucose in a tumoral cell. Warburg effect or aerobic glycolysis involves the conversion of glucose into pyruvate and subsequently into lactate, despite cancer cells possessing a fully functional respiratory chain. Enhanced glycolytic pathway not only results in increased lactate and proton secretion to the extracellular compartment but also increases the availability of glycolysis intermediates. This fact allows the increased flow of substrates to the pentose phosphate pathway (PPP) and one-carbon cycle, promoting nucleotide and lipid synthesis and maintaining redox homeostasis. (1,3-biPG: 1,3-biphosphoglycerate; 2-PG: 2-phosphoglicerate; 3-PG: 3-phosphoglicerate; ADP: adenosine diphosphate; aKG: alpha-ketoglutarate; ALDO: aldolase; ATP: adenosine triphosphate; DHAP: dihydroxyacetone phosphate; ENO: enolase; FAD^+^: flavin adenine dinucleotide; G6PD: glucose-6-phosphate dehydrogenase; GA3P: glyceraldehyde 3-phosphate; GAPDH: glyceraldehyde 3-phosphate; GDP: Guanosine diphosphate; GLUT1: glucose transporter 1; GSH: glutathione; GTP: Guanosine triphosphate; HCy: homocysteine; HK: hexokinase, LDH: lactate dehydrogenase; MCT4: monocarboxylate transporter 4; Met: methionine; NAD+: nicotinamide adenine dinucleotide; NADP: nicotinamide adenine dinucleotide phosphate; OAA: oxaloacetic acid; PEP: phosphoenolpyruvate; PFK: phosphofructokinase; PGAM: phosphoglycerate mutase; PGI: phosphoglucoisomerase; PGK: phosphoglycerate kinase; PK: pyruvate kinase; SucCoa: succinyl-coA; THF: Tetrahydrofolate).

**Figure 2 ijms-25-03142-f002:**
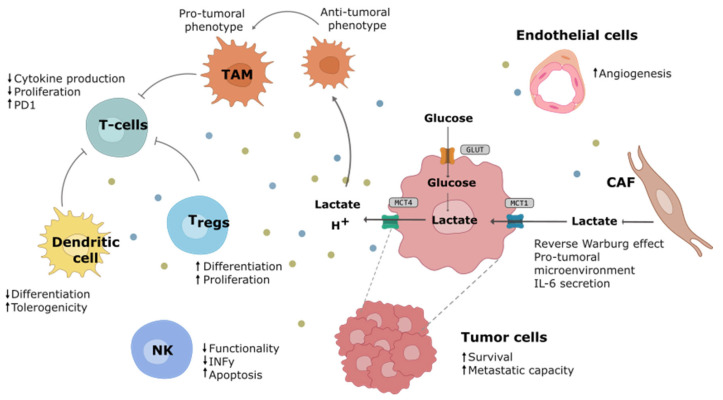
The impact of the Warburg effect on the tumor microenvironment. Tumor microenvironment (TME) is composed of different cell types, including tumor, stromal, and immune cells. Increased lactate secretion and acidification remodel these TME populations in favor of tumor progression, angiogenesis, and immunosuppression. Tumor cells and cancer-associated fibroblasts (CAFs) secrete lactate into the media, which in turn can be used by the tumor to meet energy and intermediate product requirements. This phenomenon is known as the reverse Warburg effect. Lactate and acidosis have been described to modulate the phenotype and functionality of several components of the innate and adaptative immune system, inhibiting the proliferation and cytotoxic activity of T-cells and natural killer (NK) cells as well as reducing the differentiation of dendritic cells. In contrast, regulatory T-cells (Tregs) are less sensitive to high lactate concentrations and can maintain their immunosuppressive role. Furthermore, lactate promotes the polarization of tumor-associated macrophages (TAMs) towards a pro-tumoral phenotype, thereby promoting tumor growth and invasion. (GLUT: glucose transporter; IL6: interleukine-6; INF-y: interferon-gamma; MCT1/4: monocarboxylate transporter 1/4; PD1: programmed cell death protein 1).

**Table 1 ijms-25-03142-t001:** Selected drugs affecting glycolytic enzymes are directly associated with the Warburg effect.

Target	Drug	Phase	Reference
GLUT	Glutor	Cells	[59]
	BAY-876	Pre-clinical (mice)	[60]
	KL-11743	Pre-clinical (mice)	[61]
HK	3-Bromopyruvate	Case reports	[62,63]
	2-Deoxyglucose *	Humans **	[64]
GAPDH	Koningic acid	Pre-clinical (mice)	[65]
TPI	Rabeprazole	Pre-clinical (mice)	[66]
PGMA1	PGMI-004A	Pre-clinical (mice)	[67]
	HKB99	Pre-clinical (mice)	[68]
PFK	3PO	Pre-clinical (mice)	[69]
	PFK15	Pre-clinical (mice)	[70]
	PFK158	Pre-clinical (mice)	[71]
		Phase I	[72]
LDH	FX11	Pre-clinical (mice)	[73]
	NHI-Glc-2	Pre-clinical (mice)	[74]
	NCI-006	Pre-clinical (mice)	[75]
MCTs	AZD3965	Phase I	[76]
	VB124	Pre-clinical (mice)	[77]

* 2-Deoxyglucose inhibits HK due to the accumulation of the reaction product 2-Deoxyglucose-phosphate. ** This very early clinical trial was conducted on 5 patients, equivalent to a phase I clinical trial today. GAPDH: glyceraldehyde 3-phosphate dehydrogenase; GLUT: glucose transporter 1; HK: hexokinase, LDH: lactate dehydrogenase; MCT: monocarboxylate transporter; PGMA1: phosphoglycerate mutase; PFK: phosphofructokinase; TPI: triose phosphate isomerase.

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
