# Peer review of "Targeting the Warburg Effect in Cancer: Where Do We Stand?"

_ijms, 2024, doi:10.3390/ijms25063142_

Round 1
Reviewer 1 Report
Comments and Suggestions for Authors
In this review manuscript on the topic “Targeting the Warburg effect in Cancer: Where we stand?” the authors have comprehensively analyzed the recent developments in the field of targeting tumor metabolism. In the first part of the review, authors outlined and discussed the various strategies that have been employed in various cancers that target the Warburg effect; in the second part, the authors focused on the implication of efficacy of the Warburg effect in diagnosis and the limitations of using inhibitors of glycolytic enzymes in Warburg effect in clinical settings. This review has provided knowledge on the utility and limitations of targeting the Warburg effect. The reviewer suggests the following to make the review more up-to-date.
1. The reviewer suggests the authors add in Table 1 the preclinical and clinical studies on other enzymes (part of the Warburg effect), such as HMGCS2, SDH, PDH, GAPDH, TPI, PGAM1, etc., to give due respect to the pioneers in this field.
2. The authors are advised to give a briefing on how the drugs affecting glycolytic enzymes directly associated with the Warburg effect modulate the tumor microenvironment. This information will help future studies analyze the tumor microenvironment and the possible mechanism of extrinsic resistance to these drugs.
3. The authors are suggested to include a paragraph on how the Warburg effect contributes to resistance to chemo-, radio- and immuno-therapy. This information informs the readers of the importance of targeting the Warburg effect and conventional treatments.
4. The reviewer suggests the authors include a schematic representation of how the Warburg effect influences or modulates the tumor microenvironment, especially fibroblasts, endothelial, and immune cells.
5. The authors need to explain how the Warburg effect drives cachexia states.
Comments on the Quality of English Language1. The typos have to be edited in the manuscript and the figures.
Author Response
We would like to thank the reviewer for the time and effort devoted to our manuscript. We value his/her comments as they clearly make our manuscript better and we have addressed their concerns and we have addressed their concerns in the revised version of the manuscript.
In this review manuscript on the topic “Targeting the Warburg effect in Cancer: Where we stand?” the authors have comprehensively analyzed the recent developments in the field of targeting tumor metabolism. In the first part of the review, authors outlined and discussed the various strategies that have been employed in various cancers that target the Warburg effect; in the second part, the authors focused on the implication of efficacy of the Warburg effect in diagnosis and the limitations of using inhibitors of glycolytic enzymes in Warburg effect in clinical settings. This review has provided knowledge on the utility and limitations of targeting the Warburg effect. The reviewer suggests the following to make the review more up-to-date.
The reviewer suggests the authors add in Table 1 the preclinical and clinical studies on other enzymes (part of the Warburg effect), such as HMGCS2, SDH, PDH, GAPDH, TPI, PGAM1, etc., to give due respect to the pioneers in this field.
We have followed the reviewer’s advice and incorporated some new sections in the revised version of the manuscript. New sections for GAPDH, TPI, PGAM1 are now in the text and also references included in Table 1.
The authors are advised to give a briefing on how the drugs affecting glycolytic enzymes directly associated with the Warburg effect modulate the tumor microenvironment. This information will help future studies analyze the tumor microenvironment and the possible mechanism of extrinsic resistance to these drugs.
This information was included int the future perspectives section. In the revised version of the manuscript, we have added more information and references in order to make the information more conspicuous as suggested by the reviewer.
The authors are suggested to include a paragraph on how the Warburg effect contributes to resistance to chemo-, radio- and immuno-therapy. This information informs the readers of the importance of targeting the Warburg effect and conventional treatments.
In the revised version of the manuscript, we have added some sentences according to the suggestion by the reviewer, Future perspectives section, end of first paragraph.
The reviewer suggests the authors include a schematic representation of how the Warburg effect influences or modulates the tumor microenvironment, especially fibroblasts, endothelial, and immune cells.
Following the suggestion of the reviewer, we have incorporated a new figure, Figure 2 in the revised version of the manuscript.
The authors need to explain how the Warburg effect drives cachexia states.
Following the reviewer’s suggestion we have added a paragraph in which we describe that the Warburg effect is associated with cachexia and that energy expenditure alone trough the Warburg effect is not able to explain the effects of cachexia. Also, we provide references to a couple of reviews dealing with this specific topic.
Reviewer 2 Report
Comments and Suggestions for Authors
The authors, Barba, Carrillo-Bosch and Seoane, have presented their review on the state of the field regarding research and targeting of the Warburg Effect in cancers. The premise and the title of the review are interesting although I feel the authors lack significant detail throughout, while simultaneously ignoring aspects of research presented in the literature (a few points highlighted below). It is for this reason that I feel the review requires major additions for depth and context before it can be accepted for publication.
So some of the points I feel are either not entirely scientifically accurate or are lacking detail are
- "we know metabolic pathways are a consequence and not the drivers of tumorigenesis" - this is an inaccurate statement. There has been evidence presented in the literature where metabolic dysfunction has been shown to drive tumorigenesis in multiple contexts. Indeed there is a whole sub-field of cancer research which considers caners as a metabolic disease rather than a genetic disease. As such, such a statement is not backed by scientific evidence although it is presented as unequivocal fact. Furthermore, lines 65-68 describe metabolic alterations are not the cause but a consequence. Again, this is scientifically incorrect. Take a detailed look at the work of Thomas Seyfried and others where metabolic dysfunction has been shown to be a precursor to all known cancer hallmarks. With such evidence presented in the literature, one cannot state as fact that metabolic dysfunction is simply a consequence of tumorigenesis.
- the warburg effect has been observed in most if not all cancers, not just those listed in introductory paragraph.
- line 68-69, the genes mentioned, myc, kras, wnt, these genes are highlighted as being the inducers of metabolic change. I advise the authors to read up on retrograde signalling. Retrograde signalling is the mechanism by which cells detect metabolic dysfunction and signal to the cell to activate compensatory mechanisms. In eukaryotes, the paralog of the involved gene(s) is Myc. ie, once metabolic dysfunction is detected, myc (and the other genes noted) are upregulated consequent to metabolic dysfunction detection, not prior to it.
- sections 3 and 4 are severely lacking in details and depth in both content and adequate summary of the state of the literature.
Comments on the Quality of English Language
Editing of the English is required with some phrases made not being entirely accurate, and punctuation in the wrong places. Also, a paragraph is a minimum of 2 sentences. As such certain aspects of the manuscript need to be adjusted to have proper grammatical format.
Author Response
Comments to Reviewer 2
We would like to thank the reviewer for the time and effort devoted to our manuscript. We value his/her comments as they clearly make our manuscript better and we have addressed their concerns and we have addressed their concerns in the revised version of the manuscript. In particular we have added more depth in some sections the reviewer suggested.
The authors, Barba, Carrillo-Bosch and Seoane, have presented their review on the state of the field regarding research and targeting of the Warburg Effect in cancers. The premise and the title of the review are interesting although I feel the authors lack significant detail throughout, while simultaneously ignoring aspects of research presented in the literature (a few points highlighted below). It is for this reason that I feel the review requires major additions for depth and context before it can be accepted for publication.
So some of the points I feel are either not entirely scientifically accurate or are lacking detail are
- "we know metabolic pathways are a consequence and not the drivers of tumorigenesis" - this is an inaccurate statement. There has been evidence presented in the literature where metabolic dysfunction has been shown to drive tumorigenesis in multiple contexts. Indeed there is a whole sub-field of cancer research which considers caners as a metabolic disease rather than a genetic disease. As such, such a statement is not backed by scientific evidence although it is presented as unequivocal fact. Furthermore, lines 65-68 describe metabolic alterations are not the cause but a consequence. Again, this is scientifically incorrect. Take a detailed look at the work of Thomas Seyfried and others where metabolic dysfunction has been shown to be a precursor to all known cancer hallmarks. With such evidence presented in the literature, one cannot state as fact that metabolic dysfunction is simply a consequence of tumorigenesis.
We have changed the revised version of the manuscript in order to incorporate the fact that mitochondria have the ability to sense “mitochondrial stress” and send signals to the nucleus in the mitochondrial reverse signalling and that this signalling could be prior to gene dysfunction. This is now incorporated in the last paragraph of the glycolysis section in the revised version of the manuscript.
- the warburg effect has been observed in most if not all cancers, not just those listed in introductory paragraph.
The reviewer is correct that the Warburg effect has been observed in most cancer types, however not all individual cancers show the Warburg effect; we have changed the wording of the manuscript to avoid any confusion.
- line 68-69, the genes mentioned, myc, kras, wnt, these genes are highlighted as being the inducers of metabolic change. I advise the authors to read up on retrograde signalling. Retrograde signalling is the mechanism by which cells detect metabolic dysfunction and signal to the cell to activate compensatory mechanisms. In eukaryotes, the paralog of the involved gene(s) is Myc. ie, once metabolic dysfunction is detected, myc (and the other genes noted) are upregulated consequent to metabolic dysfunction detection, not prior to it.
Please see first answer.
- sections 3 and 4 are severely lacking in details and depth in both content and adequate summary of the state of the literature.
As suggested by the reviewer we have now expanded various sections of the manuscript.
Comments on the Quality of English Language
Editing of the English is required with some phrases made not being entirely accurate, and punctuation in the wrong places. Also, a paragraph is a minimum of 2 sentences. As such certain aspects of the manuscript need to be adjusted to have proper grammatical format
In the revised version of the manuscript we have changed some aspects in order to improve grammatical format.
Round 2
Reviewer 1 Report
Comments and Suggestions for Authors
The authors have addressed all the reviewer's comments, thus making this review article comprehensive and informative.
Author Response
We would like to thank the reviewer for the time and effort devoted to the reviewing process. His/her comments have clearly improved the overall quality of the manuscript
This review paper has addressed several issues through the review process. However, I would like to point out some fundamental problems overlooked in the writing of the review paper. Please correct the following points accordingly.
However, unlike other submitted papers, this manuscript lacks numbering at the beginning of each line, making it difficult to pinpoint the exact locations that need correction. Therefore, the authors are encouraged to thoroughly revise the entire manuscript, considering the following points. Crafting eloquent expressions to present excellent content can also be a rewarding experience.
[Major concerns]
1. Abbreviations: The use of abbreviations when writing a review paper has many advantages besides simplicity of expression. To use an abbreviation, first write the abbreviation in parentheses after the full name, and then use the abbreviation from Introduction to the final Conclusion. Only in Abstract and Figure legend do it separately. If an abbreviation is not used more than twice, there is no need to define it, so please delete it.
In the revised version of the manuscript, we have checked and changed some abbreviations including, but not limited to Hexokinase (HK), lactate dehydrogenase (LDH), tumour microenvironment (TME) and pentose phosphate pathway (PPP). Some abbreviations have been deleted, for example TCA.
In cases where abbreviations are used within figures or tables, please list these abbreviations along with their corresponding full names in the figure legends or at the bottom of corresponding tables. If there are two or more abbreviations, arrange them in alphabetical order.
An abbreviation list has been added at the end of the figure legends/tables as suggested by the reviewer.
Notations of genes: In accordance with international conventions, human genes should be italicized. Therefore, please locate and correct all instances where they have been incorrectly formatted. Examples: MYC, KRAS, Wnt, etc.
We have made the changes as requested by the reviewer
Notation of chemical names: When writing chemical compound names, it is customary to use lowercase letters within the body of the text. It is not appropriate to capitalize the first letter of a chemical compound's name when it is not a proper noun, such as a brand name. Examples: Hexokinase, Pentose Phosphate Pathway, 3-Bromopyruvate, Koningic acid, and more.
We have checked and removed capital letters throughout the revised version of the manuscript as suggested by the reviewer.
Nomenclature of certain terms: Please correct instances whre the same term is spelled differently within this paper. Examples: ‘Cory Cycle’ vs. ‘Cori cycle’, etc.
We have corrected differences in spelling in the revised version of the manuscript.
[Minor concerns]
1. Define TCA.
We have now defined TCA as tricarboxylic acid cycle
2. Define DOG.
This was already defined, first sentence, third paragraph of the HK section.
3. ‘GLUT 1’ should be written as ‘GLUT1’.
We have now changed one misspelled GLUT 1 to GLUT1
4. Define PET and CT.
PET was already defined, first paragraph of the “Cancer detection through the Warburg effect” section. In the same paragraph we have defined CT.
References: It is evident that the referencing style of the cited sources does not conform to the format required by IJMS. However, the following references are missing page numbers, so please verify them again, including the page numbers. For those with page numbers starting with the Arabic numeral 1, please note that it may not represent the actual page number, so find and correct the accurate page numbers.
We would like to thank the reviewer for alerting of this error. We have now checked the references and added/changed page numbers and the format. Changes can be seen in blue in the revised version of the manuscript.
Overall, the manuscript can be considered to publication after minor revision as indicated above.
